# Fractal Contact Mechanics Model for the Rough Surface of a Beveloid Gear with Elliptical Asperities

Guangbin Yu [1,2], Hancheng Mao [1,2,]*, Lidong Jiang [3], Wei Liu [4] and Tupolev Valerii [2,4]

1   School of Mechatronics Engineering, Harbin Institute of Technology, Harbin 150001, China; yugbhit@hrbust.edu.cn
2   School of Mechanical and Power Engineering, Harbin University of Science and Technology, Harbin 150080, China; dynatupolev@outlook.com
3   CSIC No.703 Research Institute, Harbin 150078, China; csic_jiang@outlook.com
4   Jiangsu Shengan Transmission Co., Ltd., Yancheng 224000, China; liuwei@hrbeu.edu.cn
*   Correspondence: mao_hancheng@163.com

**Abstract:** Understanding the contact mechanics of rough tooth surfaces is critical in order to understand phenomena such as tooth surface flash temperature, tooth surface wear, and gear vibration. In this paper, the contact behavior between the meshing tooth surfaces of beveloid gear pairs with elliptical asperities is the focus. The contact area distribution function of the elliptical asperity was proposed for the point contact of curved surfaces by transforming the elastic contact problem between gear meshing surfaces into the contact between elastic curved surfaces with an arbitrary radius of curvature. In addition, a fractal contact mechanics model for the rough surface of a beveloid gear with elliptical asperities was established. The influence of tooth surface topography on the contact load and contact stiffness under different fractal parameters was investigated, and the results demonstrated that the real contact load and the contact stiffness of curved surfaces increase with the increase in the fractal dimension $D$ and the contact coefficient $\lambda$. Conversely, the real contact load and normal contact stiffness decrease with the increase in the fractal roughness $G$ and eccentricity $e$.

**Keywords:** mechanics model; surface topography; elliptical asperity; beveloid gears

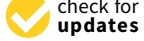



## 1. Introduction

The contact tooth surfaces are microscopically rough due to problems in machining accuracy and tooth surface wear. The contact between two rough tooth surfaces is a range of elliptical asperities of different sizes, and the distribution law of the elliptical asperities has a certain degree of anisotropy. Understanding the contact mechanics of rough tooth surfaces is critical for understanding phenomena such as tooth surface flash temperature, tooth surface wear, and gear vibration. Therefore, establishing a contact mechanics model for rough surfaces accurately and investigating the relationship between the real contact area and the contact load are particularly important.

Numerous studies on the contact analysis of rough surfaces have been conducted by domestic and foreign scholars. Greenwood and Williamson introduced the famous GW contact model and investigated the elastic contact between two rough surfaces by assuming a flat surface in contact with a rough surface [1]. Majumdar and Bhushan produced the MB contact model that considered isotropic rough surfaces to eliminate the scale-dependency of statistical theories [2]. Moreover, the normal contact load and contact area in the MB contact model were investigated using scale-independent parameters [3]. Chang established the CEB contact model considering the elastic–plastic deformation of the asperities [4]. Zhao suggested an elastic–plastic asperity contact model, namely, the ZMC model with the continuity of variables across different deformation regimes [5]. Kogut and Etsion constructed the KE contact model between an elastic hemisphere and a rigid flat surface utilizing finite element methods and analyzed the contact behavior by quantitatively separating the ranges

of elastic deformation, elastic–plastic deformation, and plastic deformation [6,7]. Jackson presented a multi-scale contact model by employing the Fourier series coefficients obtained from an FFT of the rough surfaces to describe the surface geometry at multiple scales [8,9]. Sun proposed a revised contact mechanics model for rough surfaces based on the premise that the initial profile of the rough surface should be indifferent to the sample duration and contact area. Additionally, the effects of fractal dimension, fractal roughness, and contact pressure on contact stiffness were addressed in detail [10]. Morag provided a novel elastoplastic contact model for rough surfaces that addressed a critical drawback of the MB model, which assumed that the asperities must be deformed [11]. Yuan introduced a new MB model of elastic–plastic contact between rough surfaces, and the revised model revealed that asperity levels have an effect on the mechanical characteristics of rough surfaces [12,13]. Yu established the contact mechanics model for rough surfaces based on shoulder–shoulder contact and fractal characteristics [14]. Pan constructed an analytical model for contact stiffness that incorporated the friction factor between rough contact surfaces, and the findings indicated that the friction factor has a significant effect on the contact stiffness of the whole structure [15,16]. Zhou suggested a new fractal contact model that took asperity interactions into account by representing the impacts of asperity interactions using the displacement of the mean of asperity heights [17]. Wang suggested a contact stiffness model that took asperity interactions into account, and the proposed model's correctness was proved by experiments [18]. Li developed a novel contact stiffness model for the mechanical joint surface that accounted for the asperity's continuous smooth contact features while also taking the asperity interaction into account [19]. Cohen introduced an elastic–plastic spherical contact model for combined normal and tangential loadings, which integrated previously accurate finite element analyses for contact and static friction on rough surfaces [20,21]. Wang suggested a normal contact model that took into account the contribution of elastically, plastically, and mixed elastic–plastically deformed asperities to the total normal load of rough surfaces [22]. Xiao created a novel elastoplastic asperity contact model that incorporated the continuity and smoothness of mean contact pressure and load values throughout a range of deformation regimes, from elastic to elastoplastic, and from elastoplastic to completely plastic [23]. Yu suggested a new contact stiffness model for curved surfaces that incorporated friction and was based on the continuity of the asperity length scale [24].

In recent years, the contact behavior of elliptical asperity on rough surfaces has been widely studied. Asperities on the workpiece surface may have different curvature radii in various directions, according to Kragelskii and Mikhin [25]. Horng suggested an elastic–plastic contact model for elliptic contact spots between anisotropic rough surfaces [26]. Jamari presented a theoretical model for the elastic–plastic contact of ellipsoid bodies and introduced a new simple method to analyze the complexity of the elliptical integral by an accurate approximation [27]. Jeng and Wang developed an elastic–plastic contact model that considered the elliptical contact of surface asperities [28]. To address the weaknesses of the previous elastic–plastic micro-contact model, Wen proposed a new elliptical contact model that took into account elastic–plastic deformation [29]. Jamari and Schipper experimented to obtain data on the relationship between the normal deformation of asperity and contact area and contact load and then used the fitting approach to derive the formula [30]. Lan devised an elastic–plastic contact model with ellipsoid surfaces taking friction factors into account, and he evaluated the contact load variation law for ellipsoid elastic bodies under low-velocity impact situations [31]. The research related to the contact analysis of rough curved surfaces such as tooth surfaces. Chen described the fractal approach for evaluating contact stiffness for spheroidal contact bodies with friction [32]. Liu proposed a contact model for determining the contact state of spherical pairs, taking into account the microscopic properties of the rough spherical surface and the friction factor [33]. Wang developed the contact model of the loading–unloading process for cylindrical contact surfaces with friction in different deformation stages [34]. Yang proposed the contact stiffness model of an involute arc cylindrical gear, considering the influence of sliding

friction [35]. Some scholars developed a fractal contact model applicable to a gear pair contact, considering the effect of tooth surface roughness on normal contact stiffness [36,37].

As demonstrated by the above literature review, while significant work has been conducted by domestic and foreign scholars to investigate the contact problems between rough surfaces, the current research on the fractal contact model between two rough curved surfaces has generally ignored the influence of surface texture on rough surface contact behavior. However, the topography of rough surfaces varies in the cutting direction and vertical direction, and the contact between two rough surfaces consists of a spectrum of elliptical asperities of varying diameters, with an anisotropic distribution of the elliptical asperities. Therefore, the contact area distribution function of elliptical asperity was proposed for the point contact of curved surfaces by transforming the elastic contact problem between gear meshing surfaces into the contact between elastic curved surfaces with an arbitrary radius of curvature. In addition, a fractal contact mechanics model for the rough surface of a beveloid gear with elliptical asperities was established. The influence of tooth surface topography on the contact load and contact stiffness under different fractal parameters was investigated.

## 2. A New Fractal Characterization Approach for a Rough Surface Texture

Ausloos and Berman [38] established a three-dimensional fractal function model for rough surfaces in 1985 by improving the WM function and demonstrating that the fractal function model still satisfies randomness, multiscale, and self-similarity, and the expression of the three-dimensional fractal function model is:

$$
\begin{aligned}
z(x,y) = L\left(\frac{G}{L}\right)^{(D-2)} \left(\frac{\ln g}{M}\right)^{1/2} \sum_{m=1}^{M} \sum_{n=n_{\min}}^{n_{\max}} g^{(D-3)n} \\
\times \left\{ \cos j_{m,n} - \cos[(2pg^n f(x,y)/L)\cos(g(x,y) - a_m) + j_{m,n}] \right\}
\end{aligned}
\tag{1}
$$

where, $z(x,y)$ is the three-dimensional morphological height of the rough surface. $D$ is the fractal dimension. $G$ is the fractal roughness. $n$ is the frequency index, $n = \ln l/\ln\gamma$, $l = 1/\gamma_n$. $\varphi_{m,n}$ is the random phase. $\gamma_n$ is the spatial frequency of the rough surface. $M$ indicates the number of overlapping elevated parts of the rough surface. When $M = 1$, the rough surface morphological characteristics show isotropic, when $M \neq 1$, the rough surface morphological characteristics show anisotropic. $\alpha_m$ is the three-dimensional rough surface cosine direction, $\alpha_m = \pi m/M$. $f(x,y)$ and $g(x,y)$ are the functions of independent variables related to the morphological characteristics of the rough surface, respectively.

Roughness measurements, as seen in Figure 1a, showed that the asperities dispersed on the tooth surface may be approximated by an equivalent ellipsoid and that the distribution law of the elliptical asperity on the rough tooth surface exhibits some anisotropy, as seen in Figure 1b. To this purpose, this paper improves the W-M fractal function by making $M = 1$ and $m = 1$ in Equation (1), then the dual–autonomous W-M fractal function will degenerate to a single–autonomous W-M fractal function, whose expression can be expressed as:

$$
z(x) = L\left(\frac{G}{L}\right)^{(D-1)} (\ln\gamma)^{1/2} \sum_{n=n_{\min}}^{n_{\max}} \cos\varphi_{1,n} - \cos(2\pi\gamma^n/L - \varphi_{1,n})\gamma^{(D-2)n}
\tag{2}
$$

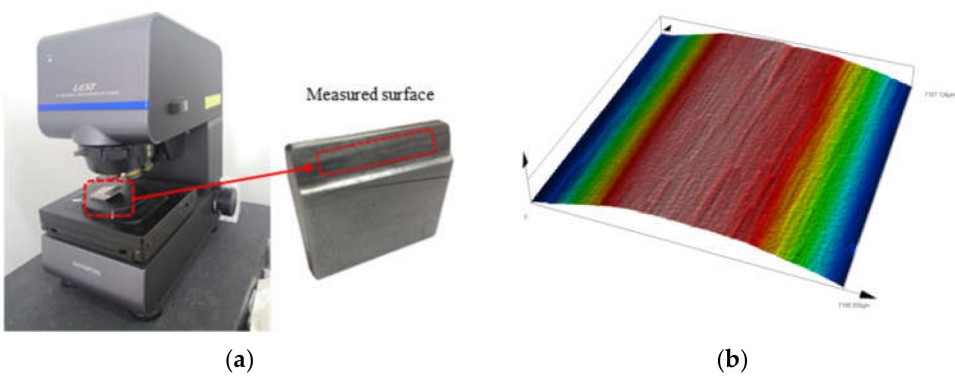

**Figure 1.** Roughness measurements of a beveloid gear tooth surface. (**a**) The measurement experiment of tooth surface topography; (**b**) The three-dimensional morphological characteristics of the rough surface.

Therefore, the fractal curves of the tooth surface topography in the cutting direction and vertical direction are superimposed to obtain a fractal description of the characteristics of the rough surface texture, whose expression can be expressed as:

$$
\begin{aligned}
z(x,y) = {} & L_1\left(\frac{G_1}{L_1}\right)^{(D_1-1)}(\ln\gamma_1)^{1/2}\sum_{n_1=n_{\min}}^{n_{\max}}\left[\cos\varphi_{1,n_1} - \cos\left(2\pi\gamma_1^n/L_1 - \varphi_{1,n}\right)\gamma^{(D_1-2)n_1}\right] \\
& + L_2\left(\frac{G_2}{L_2}\right)^{(D_2-1)}(\ln\gamma_2)^{1/2}\sum_{n_2=n_{\min}}^{n_{\max}}\left[\cos\varphi_{1,n_2} - \cos\left(2\pi\gamma_2^n/L_2 - \varphi_{1,n_2}\right)\gamma^{(D_2-2)n_2}\right]
\end{aligned}
\tag{3}
$$

where $z(x,y)$ is the height of the elliptical asperity. $D_x$, $D_y$ are the fractal dimensions in the direction $x$, $y$, $G_x$, $G_y$ are the fractal roughness in the direction $x$, $y$, respectively. $n_1$, $n_2$ are the number of sampling points within a finite length of the rough surface in the direction $x$, $y$, respectively.

By adjusting the fractal dimensions $D_x$ and $D_y$ and the fractal roughness $G_x$ and $G_y$ in the fractal characterization method (Equation (3)), the fractal characterization of the rough surface texture can be obtained, as illustrated in Figure 2a,b; it is not difficult to discover that a rough surface with a certain texture may be efficiently simulated by modifying the fractal parameters in the $x$ and $y$ dimensions.

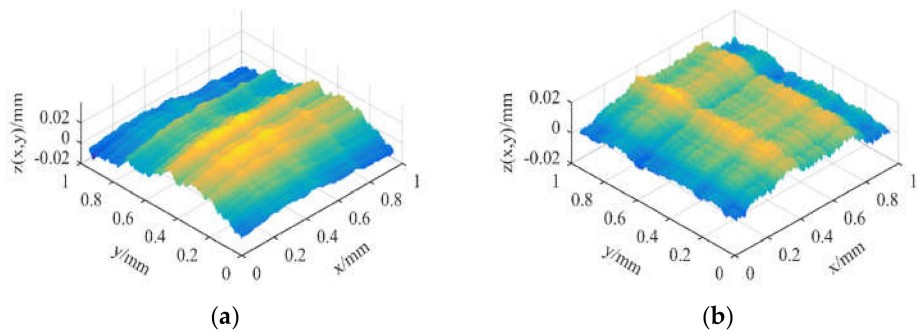

**Figure 2.** Fractal characterization of rough surfaces with different processing texture features: (**a**) $G_x = 1.0 \times 10^{-11}$ m, $G_y = 1.0 \times 10^{-9}$ m, $D_x = 1.6$, $D_y = 1.3$; (**b**) $G_x = 1.0 \times 10^{-9}$ m, $G_y = 1.0 \times 10^{-11}$ m, $D_x = 1.3$, $D_y = 1.6$.

## 3. Contact Mechanics Model of Elliptical Asperity with Rough Tooth Surface

### 3.1. Geometric Model of Single Elliptical Asperity

Since the contact between the meshing surfaces is mostly manifested as a multipoint contact, the real contact area of the tooth surface is significantly smaller than the theoretical contact area, and the contact issue between two rough surfaces is usually considered as the contact between a rigid plane and a rough surface. As shown in Figure 3, the contact model

of a single elliptical asperity was established in this paper; $p$ is the normal load applied to the elliptical asperity by a rigid plane, $\omega_n$ is the actual deformation of the elliptical asperity, and its value is between 0 and $\delta$. The profile curve $z(x,y)$ of a single elliptical asperity with the major diameter $l_x$ and the minor diameter $l_y$ of the elliptical region as the base can be deduced from Equation (3):

$$z(x,y) = (\ln \gamma_1)^{1/2} G_1{}^{D_1-1} l_x{}^{2-D_1} \cos\left(\frac{2\pi x}{l_x}\right) + (\ln \gamma_2)^{1/2} G_2{}^{D_2-1} l_y{}^{2-D_2} \cos\left(\frac{2\pi y}{l_y}\right) \quad (4)$$

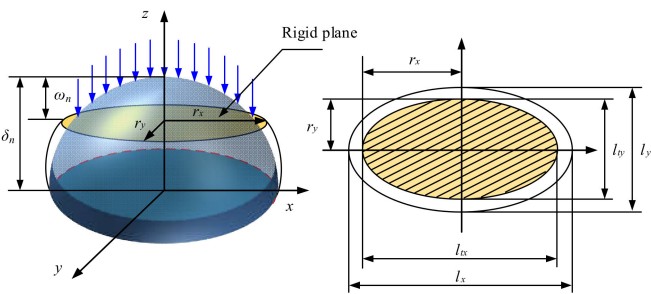

**Figure 3.** The contact model between a rigid plane and elliptical asperity.

The distance $\delta_n$ between the tip of the elliptical asperity and the base can be presented as:

$$\delta_n = z(x,y)|_{x=0,y=0} = (\ln \gamma_1)^{1/2} G_1{}^{D_1-1} l_x{}^{2-D_1} + (\ln \gamma_2)^{1/2} G_2{}^{D_2-1} l_y{}^{2-D_2} \quad (5)$$

The normal deformation $\omega_n$ of the elliptical asperity can be expressed as:

$$\omega_n = 2(\ln \gamma_1)^{1/2} G_1^{(D_1-1)} (2r_x)^{(2-D_1)} = 2(\ln \gamma_2)^{1/2} G_2^{(D_2-1)} (2r_y)^{(2-D_2)} \quad (6)$$

The geometric relationships between the normal deformation of the elliptical asperity and the effective radii of curvature are illustrated in Figure 4 and the expressions are described in Equations (7) and (8):

$$(R_x - \omega_n)^2 + r_x^2 = R_x^2 \quad (7)$$

$$(R_y - \omega_n)^2 + r_y^2 = R_y^2 \quad (8)$$

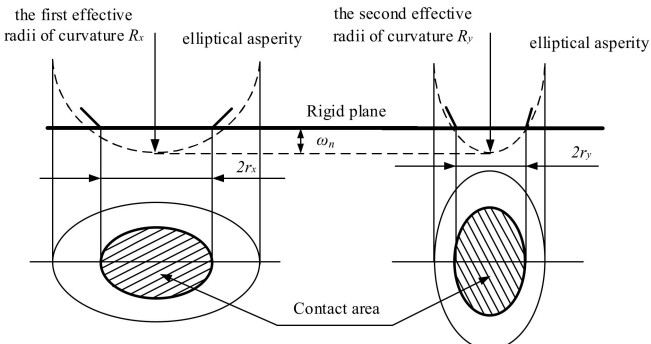

**Figure 4.** The geometric model of a single elliptical asperity.

The semiminor and semimajor radius of the elliptical asperity could well be evaluated also as the normal deformation $\omega_n$ is significantly smaller than the curvature radius of the elliptical asperity.

$$2R_x \omega_n = r_x^2 \quad (9)$$

$$2R_y \omega_n = r_y^2 \quad (10)$$

The elliptical contact area can be expressed as:

$$a = \pi r_x r_y = \pi r_x{}^2 \left(1 - e^2\right)^{1/2} = \pi r_y{}^2 \left(1 - e^2\right)^{-1/2} \tag{11}$$

Combine Equations (6), (9), (10), and (11), and the semiminor and semimajor radius of the elliptical asperity can be expressed as:

$$R_x = \frac{a^{D_1/2}\left(1 - e^2\right)^{-D_1/4}}{2^{(4-D_1)}\pi^{(D_1/2)}G_1^{(D_1-1)}(\ln \gamma_1)^{1/2}} \tag{12}$$

$$R_y = \frac{a^{D_2/2}\left(1 - e^2\right)^{D_2/4}}{2^{(4-D_2)}\pi^{(D_2/2)}G_2^{(D_2-1)}(\ln \gamma_2)^{1/2}} \tag{13}$$

where $e$ is the eccentricity of the contact ellipse, which can be expressed as:

$$e^2 = 1 - \left(\frac{r_y}{r_x}\right)^2, r_y < r_x \tag{14}$$

When the eccentricity $e$ of the contact ellipse is 0, the elliptical contact area will be transformed into a circular contact area, namely, $R_x = R_y = R_m$. When the eccentricity $e$ is not 0, the equivalent radius of curvature $R_m$ corresponding to the elliptical asperity can be expressed as:

$$\frac{1}{R_m} = \frac{1}{2}\left(\frac{1}{R_x} + \frac{1}{R_y}\right) \tag{15}$$

*3.2. Contact Mechanics Model of Single Elliptical Asperity*

3.2.1. Elastic Contact of Elliptical Asperity

According to Hertz's theory, when an elliptical asperity is deformed significantly less than the critical elastic deformation, namely, $\omega_n \leq \omega_{nec}$, it merely experiences elastic contact deformation. The maximum contact pressure $P_m$, the semimajor radius of the contact ellipse $r_x$, and the elastic deformation $\omega$ can be expressed, respectively, as:

$$P_m = \frac{3F}{2a} \tag{16}$$

$$r_x = \left[\frac{3E(e)FR_m}{2\pi E'\left(1 - e^2\right)}\right]^{1/3} \tag{17}$$

$$\omega = \frac{2K(e)}{\pi}\left[\frac{\pi\left(1 - e^2\right)}{2E(e)R_m}\right]^{1/3}\left(\frac{3F}{4E'}\right)^{2/3} \tag{18}$$

where $K(e)$ and $E(e)$ are the complete elliptic integrals of the first and second kind, respectively, whose expressions are:

$$K(e) = \int_0^{\pi/2} \frac{d\varphi}{\sqrt{1 - e^2 \sin^2 \varphi}} \tag{19}$$

$$E(e) = \int_0^{\pi/2} \sqrt{1 - e^2 \sin^2 \varphi}\, d\varphi \tag{20}$$

where $E'$ is the effective elastic modulus, which can be calculated by the following equation:

$$\frac{1}{E'} = \frac{1 - v_1^2}{E_1} + \frac{1 - v_2^2}{E_2} \tag{21}$$

where $\nu_1$, $\nu_2$ and $E_1$, $E_2$ are Poisson's ratio and the elastic modulus of the two rough surface materials in contact with each other, respectively.

Combining Equations (16)–(18), the real contact area and real contact load of elastic deformation for elliptical asperity can be presented as:

$$a_{ne}(\omega) = \left[ \frac{E(e)}{K(e)(1-e^2)^{1/2}} \right] \pi R_m \omega = f_1(e)\pi R_m \omega \tag{22}$$

$$F_{ne}(\omega) = \left[ \frac{\pi E(e)^{-1/2}}{2K(e)^{3/2}(1-e^2)^{1/2}} \right] \frac{4}{3} E' R_m^{1/2} \omega^{3/2} = f_2(e)\frac{4}{3}E' R_m^{1/2}\omega^{3/2} \tag{23}$$

Therefore, the average contact pressure $P_e$ can be expressed as:

$$P_e(\omega) = \frac{F_e(\omega)}{a_e(\omega)} = \frac{4f_2(e)E'}{3f_1(e)\pi}\left(\frac{\omega}{R_m}\right)^{1/2} \tag{24}$$

The asperity entirely enters a plastic deformation state when the maximal contact pressure $P_m$ is equal to $KH$, and the equation is:

$$H = 2.8\sigma_y \tag{25}$$

where, $H$ denotes the material hardness, $K$ denotes the hardness coefficient, and the relationship between the material Poisson's ratio is:

$$K = 0.454 + 0.41\nu \tag{26}$$

The critical normal deformation of a single elliptical asperity between elastic and inelastic deformation is:

$$\omega_{nec} = K(e)E(e)R_m\left(\frac{KH}{E'}\right)^2 \tag{27}$$

By combining Equations (15), (22) and (27), the expression for the critical contact area $a_{nec}$ between elastic and inelastic deformation can be deduced as:

$$a_{nec}\left( \frac{2^{(3-D_1)}\pi^{\frac{D_1}{2}}G_1^{(D_1-1)}(\ln\gamma_1)^{\frac{1}{2}}}{(1-e^2)^{-\frac{D_1}{4}}}a_{nec}^{-\frac{D_1}{2}} + \frac{2^{(3-D_2)}\pi^{\frac{D_2}{2}}G_2^{(D_2-1)}(\ln\gamma_2)^{\frac{1}{2}}}{(1-e^2)^{\frac{D_2}{4}}}a_{nec}^{-\frac{D_2}{2}} \right)^2 = 4f_1(e)\pi K(e)E(e)\left(\frac{KH}{E'}\right)^2 \tag{28}$$

### 3.2.2. Elastic–Plastic Contact of Elliptical Asperity

As shown in Figure 5, Kogut and Etsion [7] investigated the mechanism of elastic–plastic deformation occurring in asperities on rough surfaces by performing a finite element contact analysis of a single elastic sphere with a smooth rigid plane; the results of the study showed that the asperities are in elastic–plastic deformation when the actual deformation $\omega_n$ of the asperities on the rough surface is greater than the critical elastic deformation $\omega_{nec}$ and less than or equal to 110 $\omega_{nec}$. In addition, a large number of simulation results have proven that the elastic–plastic deformation is divided into two stages: the asperities are in the first elastic–plastic deformation stage when the actual deformation $\omega_n$ is greater than the critical elastic deformation $\omega_{nec}$ and less than or equal to 6 $\omega_{nec}$, and the asperities are in the second elastic–plastic deformation stage when the actual deformation $\omega_n$ is greater than the critical elastic deformation 6 $\omega_{nec}$ and less than or equal to 110 $\omega_{nec}$. 6 $\omega_{nec}$ and 110 $\omega_{nec}$ are defined as the critical elastic–plastic deformation and critical plastic deformation, respectively, and their expressions can be expressed as follows, respectively:

$$\omega_{nepc} = 6\omega_{nec} = 6K(e)E(e)R_m\left(\frac{KH}{E'}\right)^2 \tag{29}$$

$$\omega_{npc} = 110\omega_{nec} = 110K(e)E(e)R_m\left(\frac{KH}{E'}\right)^2 \tag{30}$$

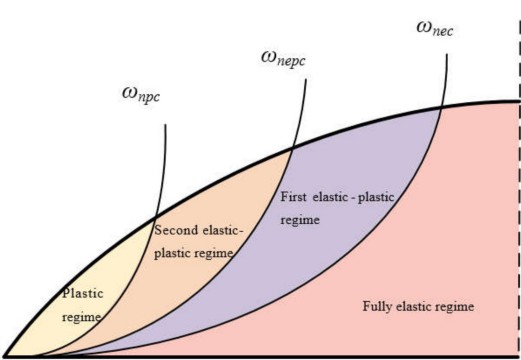

**Figure 5.** Relationship between the actual deformation $\omega_n$ of asperity and the deformation mode.

Contact mechanics demands that the contact process of the asperity should be continuous. Therefore, the elliptical asperity is in the first elastic–plastic deformation when the real contact area satisfies $a_{nec} \geq a_n \geq a_{nepc}$, and the elliptical asperity is in the second elastic–plastic deformation when the real contact area satisfies $a_{nepc} \geq a_n \geq a_{npc}$. The critical elastic–plastic contact area $a_{nepc}$ and the critical plastic contact area $a_{npc}$ can be expressed as:

$$a_{nepc}\left(\frac{2^{(3-D_1)}\pi^{\frac{D_1}{2}}G_1^{(D_1-1)}(\ln\gamma_1)^{\frac{1}{2}}}{(1-e^2)^{-\frac{D_1}{4}}}a_{nepc}^{-\frac{D_1}{2}} + \frac{2^{(3-D_2)}\pi^{\frac{D_2}{2}}G_2^{(D_2-1)}(\ln\gamma_2)^{\frac{1}{2}}}{(1-e^2)^{\frac{D_2}{4}}}a_{nepc}^{-\frac{D_2}{2}}\right)^2 = 24f_1(e)\pi K(e)E(e)\left(\frac{KH}{E'}\right)^2 \tag{31}$$

$$a_{nec}\left(\frac{2^{(3-D_1)}\pi^{\frac{D_1}{2}}G_1^{(D_1-1)}(\ln\gamma_1)^{\frac{1}{2}}}{(1-e^2)^{-\frac{D_1}{4}}}a_{nec}^{-\frac{D_1}{2}} + \frac{2^{(3-D_2)}\pi^{\frac{D_2}{2}}G_2^{(D_2-1)}(\ln\gamma_2)^{\frac{1}{2}}}{(1-e^2)^{\frac{D_2}{4}}}a_{nec}^{-\frac{D_2}{2}}\right)^2 = 4f_1(e)\pi K(e)E(e)\left(\frac{KH}{E'}\right)^2 \tag{32}$$

The expressions of the contact area and contact load when the asperity is in the first elastic–plastic deformation can be expressed as follows, respectively:

$$\frac{a_{nep1}}{a_{nec}} = m_{nep1}\left(\frac{\omega_n}{\omega_{nec}}\right)^{n_{nep1}} \tag{33}$$

$$\frac{F_{nep1}}{F_{nec}} = c_{nep1}\left(\frac{\omega_n}{\omega_{nec}}\right)^{d_{nep1}} \tag{34}$$

where $m_{ep1}$ = 0.93, $n_{ep1}$ = 1.136, $c_{ep1}$ = 1.03, $d_{ep1}$ = 1.425.

The expressions of the contact area and contact load when the asperity is in the second elastic–plastic deformation can be expressed as follows, respectively:

$$\frac{a_{nep2}}{a_{nec}} = m_{nep2}\left(\frac{\omega_n}{\omega_{nec}}\right)^{n_{nep2}} \tag{35}$$

$$\frac{F_{nep2}}{F_{nec}} = c_{nep2}\left(\frac{\omega_n}{\omega_{nec}}\right)^{d_{nep2}} \tag{36}$$

where $m_{ep2}$ = 0.94, $n_{ep2}$ = 1.146, $c_{ep2}$ = 1.40, $d_{ep2}$ = 1.263.

### 3.2.3. Plastic Contact of Elliptical Asperity

When the real contact deformation is greater than the critical plastic deformation, namely, $\omega_n \geq 110\,\omega_{nec}$, the asperity is in full plastic deformation, and the contact area $a_{np}$ and the normal contact pressure $F_{np}$ can be expressed as follows:

$$a_{np} = \pi R_m \omega_n f_1(e) \tag{37}$$

$$F_{np} = Ha_{np} \tag{38}$$

### 3.3. Modified Model of the Island Area Distribution Function for a Point Contact

### 3.3.1. Contact Area Distribution Function

Mandelbrot [39] discovered the distribution law of ocean islands in the study of earth geomorphology, and its expression is:

$$N(A > a) = \left(\frac{a_l}{a}\right)^{D/2} \tag{39}$$

Majumdar and Bhushan linked Mandelbrot's island area distribution theory to the contact area distribution of the fractal rough surface, and then proposed a functional relationship between the contact area distribution of discrete asperities and the fractal dimension, which can be expressed as follows:

$$n(a) = \frac{D}{2} \frac{a_l^{D/2}}{a^{(D/2+1)}} \tag{40}$$

By integrating the contact area distribution function shown in Equation (40), the total real contact area $A_r$ can be obtained as:

$$A_r = \int_0^{a_l} n(a)a\,da = \int_0^{a_l} \frac{D}{2} a_l^{D/2} a^{-D/2}\,da = \frac{D}{2-D} a_l \tag{41}$$

### 3.3.2. Elliptical Contact Area Distribution Function

As illustrated in Figure 6, the contact area of an elliptical asperity on the rough tooth surface is an ellipse with a semiminor radius $r_x$ and a semimajor radius $r_y$.

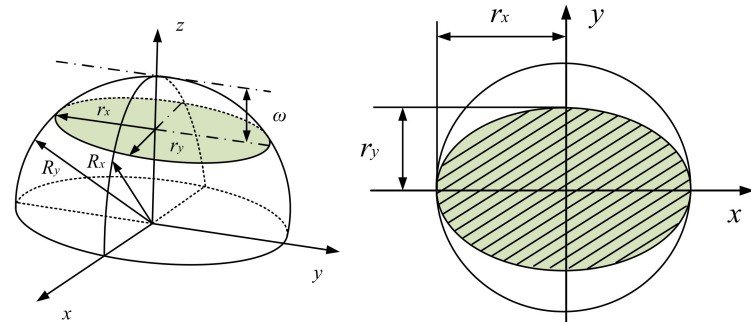

**Figure 6.** Comparison of the contact area between spherical asperity and elliptical asperity.

Based on Equation (40), the expression for the total number of islands between the contact area $a$ and the maximum contact area $a_l$ of any elliptical asperities can be deduced as:

$$N(A > a) = \left(\frac{a_l}{a}\right)^{\zeta(D_1+D_2)/2} \tag{42}$$

where $D_1$, $D_2$ are the fractal dimensions of the rough surface in the $x$-axis and $y$-axis directions, respectively. $\zeta$ is the contact coefficient of the elliptical asperity, and the contact coefficient $\zeta$ can be expressed as:

$$\zeta = \frac{1}{2}\left(1 - e^2\right)^{1/2} \tag{43}$$

Thus, the contact area distribution of elliptical asperities can be obtained as a function of the fractal dimension:

$$n(a) = \frac{\zeta(D_1 + D_2)}{2} \frac{a_l^{\zeta(D_1+D_2)/2}}{a^{(\zeta(D_1+D_2)/2+1)}} \tag{44}$$

The total contact area $A_r$ can be obtained by integrating the contact area distribution function described in Equation (44) as:

$$A_r = \int_0^{a_l} n(a)a\,da = \frac{\zeta(D_1 + D_2)}{2 - \zeta(D_1 + D_2)} a_l \tag{45}$$

### 3.3.3. Contact Coefficient of the Elliptical Area of a Beveloid Gear

As shown in Figure 7, the contact area of beveloid gear pairs is elliptical. Assuming that the principal curvatures of tooth surfaces are $\rho_{11}$, $\rho_{12}$, $\rho_{21}$, and $\rho_{22}$, respectively, the contact area distribution functions of the beveloid gear should satisfy the following relationship:

$$n'(A) = \lambda n(A) \tag{46}$$

where $\lambda$ is the contact surface coefficient, which can be expressed as:

$$\lambda = \left(\frac{S}{\sum S}\right)^{\rho_m} \tag{47}$$

where $S$ is the theoretical contact area. $\sum S$ is the sum of the surface areas of the two elastic tooth surfaces, and $\rho_m$ is the integrated curvature coefficient, whose expression is:

$$\rho_m = \rho_{11} + \rho_{12} + \rho_{21} + \rho_{22} \tag{48}$$

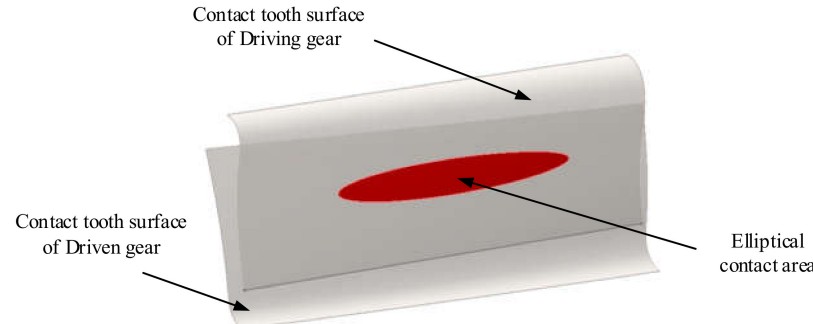

**Figure 7.** Schematic diagram of a beveloid gear with the crossed shaft.

According to Hertz's theory, the contact area will be transformed from the theoretical point contact to an approximately elliptical contact under the action of a normal contact load on the tooth surface, and the semimajor radius $r_x$ of the contact ellipse can be expressed as:

$$r_x = \left[\frac{3FE(e)}{2\pi E'(1 - e^2)\rho_m}\right]^{1/3} \tag{49}$$

where $F$ is the contact load on the tooth surface.

The elliptical contact area of the beveloid gear pair is:

$$S = \pi r_x r_y = \pi r_x^2 \sqrt{1 - e^2} = \pi \left[\frac{3FE(e)}{2\pi E'\rho_m}\right]^{2/3} \left(1 - e^2\right)^{1/6} \tag{50}$$

The sum of the surface areas of the two elastic surfaces ΣS can be expressed as:

$$\sum S = 2\pi(\rho_{n1} + \rho_{n2})L \tag{51}$$

where $L$ is the contact length of the two gears, and $\rho_{n1}$ and $\rho_{n2}$ are the radii of curvature at the nodes of the gear and pinion, respectively, in the form of:

$$\rho_{n1} = \frac{d_1 \sin \alpha_n}{2 \cos \beta}, \; \rho_{n2} = \frac{d_2 \sin \alpha_n}{2 \cos \beta} \tag{52}$$

Substituting Equations (50)–(52) into Equation (47), respectively, the contact coefficient of the elliptical contact area can be expressed as:

$$\lambda = \left\{ \frac{2\pi \left[\frac{3FE(e)}{2\pi E' \rho_m}\right]^{2/3} \cos \beta}{(d_1 + d_2) \sin \alpha_n L (1 - e^2)^{-1/6}} \right\}^{\rho_m} \tag{53}$$

Figure 8 shows the trend of the contact coefficient $\lambda$. The eccentricity $e = 0.3$, the number of teeth of the gear is 45, the number of teeth of the pinion is 29, the normal pressure angle $\alpha_n = 20°$, and the modulus $m = 4$ mm. As seen in Figure 8a, the contact coefficient reduces as the integrated curvature of the elastic surface increases. When the integrated curvature of the elastic surface tends to 0, the two elastic surfaces are approximately in-plane contact and the surface contact coefficient $\lambda$ tends to 1. As seen in Figure 8b, the contact coefficient rises when the contact load increases. This is because as the contact load increases, the contact area of the elliptical asperities grows proportionately, resulting in an increase in the surface contact coefficient.

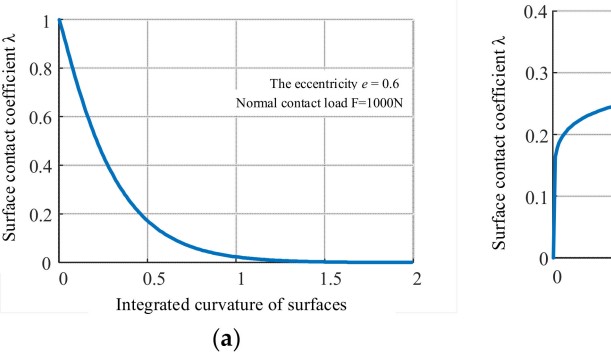
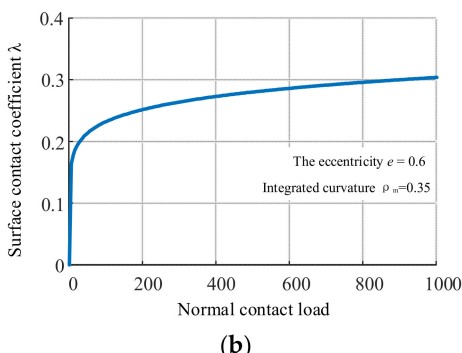

(a)　　　　　　　　　　　　　　　　　(b)

**Figure 8.** Fractal characterization of rough surfaces with different processing texture features: (a) $G_x = 1.0 \times 10^{-11}$ m, $G_y = 1.0 \times 10^{-9}$ m, $D_x = 1.6$, $D_y = 1.3$; (b) $G_x = 1.0 \times 10^{-9}$ m, $G_y = 1.0 \times 10^{-11}$ m, $D_x = 1.3$, $D_y = 1.6$.

Therefore, the total elliptical contact area can be expressed as:

$$A_r = \int_0^{a_l} \lambda n'(a) a \, da = \lambda \frac{\zeta(D_1 + D_2)}{2 - \zeta(D_1 + D_2)} a_l \tag{54}$$

## 4. The Fractal Contact Model for Rough Curved Surfaces with Elliptical Asperities

### 4.1. Real Contact Area and Contact Load

#### 4.1.1. Real Contact Area

The smallest contact area is considered to converge to 0 based on the size distribution of contact spots. When $a_l > a_{nec}$, the asperities may deform totally elastically, totally plastically, or both elastically and plastically. The real contact area $A_r$ is the sum of the plastic contact

area $A_{rp}$, the second plastic contact area is $A_{rep2}$, the first plastic contact area is $A_{rep1}$, and the elastic contact area is $A_{re}$, namely:

$$A_r = A_{rp} + A_{rep2} + A_{rep1} + A_{re} \tag{55}$$

$$A_{rp} = \int_0^{a_{npc}} n'(a)a\,da = \lambda \frac{\zeta(D_1 + D_2)}{2 - \zeta(D_1 + D_2)} a_l^{\frac{\zeta(D_1+D_2)}{2}} a_{npc}^{\frac{2-\zeta(D_1+D_2)}{2}} \tag{56}$$

$$A_{rep2} = \int_{a_{npc}}^{a_{nepc}} n'(a)a\,da = \lambda \frac{\zeta(D_1 + D_2)}{2 - \zeta(D_1 + D_2)} a_l^{\frac{\zeta(D_1+D_2)}{2}} \left( a_{nepc}^{\frac{2-\zeta(D_1+D_2)}{2}} - a_{npc}^{\frac{2-\zeta(D_1+D_2)}{2}} \right) \tag{57}$$

$$A_{rep1} = \int_{a_{nepc}}^{a_{nec}} n'(a)a\,da = \lambda \frac{\zeta(D_1 + D_2)}{2 - \zeta(D_1 + D_2)} a_l^{\frac{\zeta(D_1+D_2)}{2}} \left( a_{nec}^{\frac{2-\zeta(D_1+D_2)}{2}} - a_{nepc}^{\frac{2-\zeta(D_1+D_2)}{2}} \right) \tag{58}$$

$$A_{re} = \int_{a_{nec}}^{a_l} n'(a)a\,da = \lambda \frac{\zeta(D_1 + D_2)}{2 - \zeta(D_1 + D_2)} a_l^{\frac{\zeta(D_1+D_2)}{2}} \left( a_l^{\frac{2-\zeta(D_1+D_2)}{2}} - a_{nec}^{\frac{2-\zeta(D_1+D_2)}{2}} \right) \tag{59}$$

### 4.1.2. Real Contact Load

When $a_l > a_{nec}$, the real contact load on the rough surface can be expressed as:

$$F_r = F_{rnp} + F_{rnep2} + F_{rnep1} + F_{rne} \tag{60}$$

where the real contact loads for plastic deformation, second elastic–plastic deformation, and first elastic–plastic deformation can be expressed, respectively, as:

$$F_{rnp} = \int_0^{a_{npc}} F_{np} n'(a)\,da = K\sigma_y \lambda \frac{\zeta(D_1 + D_2)}{2 - \zeta(D_1 + D_2)} a_l^{\frac{\zeta(D_1+D_2)}{2}} a_{npc}^{\frac{2-\zeta(D_1+D_2)}{2}} \tag{61}$$

$$\begin{aligned} F_{rnep2} = \int_{a_{npc}}^{a_{nepc}} F_{nep2} n'(a)\,da = F_{nec} c_{ep2} m_{ep2}^{-\frac{d_{ep2}}{m_{ep2}}} a_{nec}^{-\frac{d_{ep2}}{n_{ep2}}} \lambda \frac{n_{ep2}[\zeta(D_1+D_2)]}{2d_{ep2} - \zeta(D_1+D_2)n_{ep2}} \\ a_l^{\zeta(D_1+D_2)/2} \left( a_{nepc}^{\frac{2d_{ep2} - \zeta(D_1+D_2)n_{ep2}}{2n_{ep2}}} - a_{npc}^{\frac{2d_{ep2} - \zeta(D_1+D_2)n_{ep2}}{2n_{ep2}}} \right) \end{aligned} \tag{62}$$

$$\begin{aligned} F_{rnep1} = \int_{a_{nepc}}^{a_{nec}} F_{nep1} n'(a)\,da = F_{nec} c_{ep1} m_{ep1}^{-\frac{d_{ep1}}{m_{ep1}}} a_{nec}^{-\frac{d_{ep1}}{n_{ep1}}} \lambda \frac{n_{ep1}[\zeta(D_1+D_2)]}{2d_{ep1} - \zeta(D_1+D_2)n_{ep1}} \\ a_l^{\zeta(D_1+D_2)/2} \left( a_{nec}^{\frac{2d_{ep1} - \zeta(D_1+D_2)n_{ep1}}{2n_{ep1}}} - a_{nepc}^{\frac{2d_{ep1} - \zeta(D_1+D_2)n_{ep1}}{2n_{ep1}}} \right) \end{aligned} \tag{63}$$

Combining Equations (15), (22), and (23), the relationship between the contact area and contact load of the elliptical asperities in the elastic deformation can be obtained as:

$$F_{ne}(a) = \frac{2f_2(e)E'}{3[f_1(e)]^{3/2}\pi^{3/2}} \left( \frac{2^{(4-D_1)}\pi^{(D_1/2)}G_1^{(D_1-1)}(\ln\gamma_1)^{1/2}}{a^{(D_1-3)/2}(1-e^2)^{D_1/4}} + \frac{2^{(4-D_2)}\pi^{(D_2/2)}G_2^{(D_2-1)}(\ln\gamma_2)^{1/2}}{a^{(D_2-3)/2}(1-e^2)^{-D_2/4}} \right) \tag{64}$$

Therefore, the real contact load of the elliptical asperities in the elastic deformation can be expressed as:

$$\begin{aligned} F_{rne} = {} & \frac{2^{(5-D_1)}\pi^{(D_1-3)/2} f_2(e)E'G_1^{(D_1-1)}(\ln\gamma_1)^{1/2}\zeta(D_1+D_2)}{3[f_1(e)]^{3/2}(1-e^2)^{1/4D_1}(3-\zeta(D_1+D_2)-D_1)} a_l^{\frac{\zeta(D_1+D_2)}{2}} \left( a_l^{\frac{3-\zeta(D_1+D_2)-D_1}{2}} - a_{nec}^{\frac{3-\zeta(D_1+D_2)-D_1}{2}} \right) \\ & + \frac{2^{(5-D_2)}\pi^{(D_2-3)/2} f_2(e)E'G_2^{(D_2-1)}(\ln\gamma_2)^{1/2}\zeta(D_1+D_2)}{3[f_1(e)]^{3/2}(1-e^2)^{1/4D_2}(3-\zeta(D_1+D_2)-D_2)} a_l^{\frac{\zeta(D_1+D_2)}{2}} \left( a_l^{\frac{3-\zeta(D_1+D_2)-D_2}{2}} - a_{nec}^{\frac{3-\zeta(D_1+D_2)-D_2}{2}} \right) \end{aligned} \tag{65}$$

### 4.1.3. The Relationship between the Real Contact Area and the Real Contact Load

When $a_l > a_{nec}$, the elliptical asperities in the first elastic–plastic deformation, the second elastic–plastic deformation, plastic deformation, and elastic deformation are significant.

$$
\begin{aligned}
F_{r1} = &\ g_1(D_1,D_2)\left[\frac{2-\zeta(D_1+D_2)}{\lambda\zeta(D_1+D_2)}A_r\right]^{\frac{\zeta(D_1+D_2)}{2}}\left(\left[\frac{2-\zeta(D_1+D_2)}{\lambda\zeta(D_1+D_2)}A_r\right]^{\frac{3-\zeta(D_1+D_2)-D_1}{2}}-a_{nec}^{\frac{3-\zeta(D_1+D_2)-D_1}{2}}\right) \\
&+g_2(D_1,D_2)\left[\frac{2-\zeta(D_1+D_2)}{\lambda\zeta(D_1+D_2)}A_r\right]^{\frac{\zeta(D_1+D_2)}{2}}\left(\left[\frac{2-\zeta(D_1+D_2)}{\lambda\zeta(D_1+D_2)}A_r\right]^{\frac{3-\zeta(D_1+D_2)-D_2}{2}}-a_{nec}^{\frac{3-\zeta(D_1+D_2)-D_2}{2}}\right) \\
&+F_{nec}c_{ep1}m_{ep1}^{-\frac{d_{ep1}}{m_{ep1}}}a_{nec}^{-\frac{d_{ep1}}{n_{ep1}}}g_3(D_1,D_2)A_r^{\frac{\zeta(D_1+D_2)}{2}}\left(a_{nec}^{\frac{2d_{ep1}-\zeta(D_1+D_2)n_{ep1}}{2n_{ep1}}}-a_{nepc}^{\frac{2d_{ep1}-\zeta(D_1+D_2)n_{ep1}}{2n_{ep1}}}\right) \\
&+F_{nec}c_{ep2}m_{ep2}^{-\frac{d_{ep2}}{m_{ep2}}}a_{nec}^{-\frac{d_{ep2}}{n_{ep2}}}g_4(D_1,D_2)A_r^{\frac{\zeta(D_1+D_2)}{2}}\left(a_{nepc}^{\frac{2d_{ep2}-\zeta(D_1+D_2)n_{ep2}}{2n_{ep2}}}-a_{npc}^{\frac{2d_{ep2}-\zeta(D_1+D_2)n_{ep2}}{2n_{ep2}}}\right) \\
&K\sigma_y g_5(D_1,D_2)A_r^{\frac{\zeta(D_1+D_2)}{2}}a_{npc}^{\frac{2-\zeta(D_1+D_2)}{2}}
\end{aligned}
\tag{66}
$$

$$
g_1(D_1,D_2) = \frac{2^{(5-D_1)}\pi^{(D_1-3)/2}f_2(e)E'G_1^{(D_1-1)}(\ln\gamma_1)^{1/2}\zeta(D_1+D_2)}{3[f_1(e)]^{3/2}(1-e^2)^{1/4D_1}(3-\zeta(D_1+D_2)-D_1)}
\tag{67}
$$

$$
g_2(D_1,D_2) = \frac{2^{(5-D_2)}\pi^{(D_2-3)/2}f_2(e)E'G_2^{(D_2-1)}(\ln\gamma_2)^{1/2}\zeta(D_1+D_2)}{3[f_1(e)]^{3/2}(1-e^2)^{1/4D_2}(3-\zeta(D_1+D_2)-D_2)}
\tag{68}
$$

$$
g_3(D_1,D_2) = \frac{\lambda n_{ep1}[\zeta(D_1+D_2)]}{2d_{ep1}-\zeta(D_1+D_2)n_{ep1}}\left[\frac{2-\zeta(D_1+D_2)}{\lambda\zeta(D_1+D_2)}\right]^{\frac{\zeta(D_1+D_2)}{2}}
\tag{69}
$$

$$
g_4(D_1,D_2) = \frac{\lambda n_{ep2}[\zeta(D_1+D_2)]}{2d_{ep2}-\zeta(D_1+D_2)n_{ep2}}\left[\frac{2-\zeta(D_1+D_2)}{\lambda\zeta(D_1+D_2)}\right]^{\frac{\zeta(D_1+D_2)}{2}}
\tag{70}
$$

$$
g_5(D_1,D_2) = \left[\frac{\lambda\zeta(D_1+D_2)}{2-\zeta(D_1+D_2)}\right]^{\frac{2-\zeta(D_1+D_2)}{2}}
\tag{71}
$$

### 4.2. Calculation of the Contact Stiffness of Rough Tooth Surfaces

4.2.1. Contact Stiffness Model of a Single Elliptical Asperity

When the contact area of the largest contact spot $a_l < a_{npc}$, the elliptical asperity is only in plastic deformation, and the normal contact stiffness $k_{np}$ can be expressed as:

$$
k = k_{np} = 0
\tag{72}
$$

When $a_{nepc} > a_l > a_{npc}$, the elliptical asperity is in the second elastic–plastic deformation, and the normal contact stiffness $k_{nep2}$ can be expressed as:

$$
k_{nep2} = \frac{dF_{nep2}}{da}\frac{da}{d\omega} = \frac{F_{nec}c_{ep2}d_{ep2}m_{ep2}^{\frac{1-d_{ep2}}{n_{ep2}}}a_{nec}^{\frac{1-d_{ep2}}{n_{ep2}}}}{\omega_{nec}}a^{\frac{d_{ep2}-1}{n_{ep2}}}
\tag{73}
$$

When $a_{nec} > a_l > a_{nep}$, the elliptical asperity is in the first elastic–plastic deformation, and the normal contact stiffness $k_{nep1}$ can be expressed as:

$$
k_{nep1} = \frac{dF_{nep1}}{da}\frac{da}{d\omega} = \frac{F_{nec}c_{ep1}d_{ep1}m_{ep1}^{\frac{1-d_{ep1}}{n_{ep1}}}a_{nec}^{\frac{1-d_{ep1}}{n_{ep1}}}}{\omega_{nec}}a^{\frac{d_{ep1}-1}{n_{ep1}}}
\tag{74}
$$

When $a_l > a_{nec}$, the elliptical asperity is only in elastic deformation, and the normal contact stiffness $k_{ne}$ can be expressed as:

$$
k_{ne} = \frac{dF_{ne}}{da}\frac{da}{d\omega} = \frac{2f_2(e)E'}{[f_1(e)]^{1/2}\pi^{1/2}}a^{1/2}
\tag{75}
$$

#### 4.2.2. Contact Stiffness Model of a Rough Surface

When $a_l > a_{nec}$, the normal contact stiffness $K$ of the rough surface can be expressed as:

$$K = K_{np} + K_{nep2} + K_{nep1} + K_{ne} \tag{76}$$

When the elliptical asperity is in the second elastic–plastic deformation, the normal contact stiffness $k_{nep2}$ can be presented as:

$$k_{nep2} = \int_{a_{npc}}^{a_{nepc}} k_{nep2} n'(a) da = \frac{F_{nec} c_{ep2} d_{ep2} m_{ep2}^{\frac{1-d_{ep2}}{n_{ep2}}} a_{nec}^{\frac{1-d_{ep2}}{n_{ep2}}}}{\omega_{nec}} \frac{n_{ep2} \zeta (D_1 + D_2) a_l^{\zeta(D_1+D_2)/2}}{2d_{ep2} - 2 - n_{ep2}[\zeta(D_1 + D_2) + 2] + 2n_{ep2}}$$

$$\left( a_{nepc}^{\frac{2d_{ep2}-2-n_{ep2}[\zeta(D_1+D_2)+2]+2n_{ep2}}{2n_{ep2}}} - a_{npc}^{\frac{2d_{ep2}-2-n_{ep2}[\zeta(D_1+D_2)+2]+2n_{ep2}}{2n_{ep2}}} \right) \tag{77}$$

When the elliptical asperity is in the first elastic–plastic deformation, the normal contact stiffness $k_{nep1}$ can be presented as:

$$k_{nep1} = \int_{a_{nepc}}^{a_{nec}} k_{nep1} n'(a) da = \frac{F_{nec} c_{ep1} d_{ep1} m_{ep1}^{\frac{1-d_{ep1}}{n_{ep1}}} a_{nec}^{\frac{1-d_{ep1}}{n_{ep1}}}}{\omega_{nec}} \frac{n_{ep2} \zeta (D_1 + D_2) a_l^{\zeta(D_1+D_2)/2}}{2d_{ep1} - 2 - n_{ep1}[\zeta(D_1 + D_2) + 2] + 2n_{ep1}}$$

$$\left( a_{nec}^{\frac{2d_{ep1}-2-n_{ep1}[\zeta(D_1+D_2)+2]+2n_{ep1}}{2n_{ep1}}} - a_{nepc}^{\frac{2d_{ep1}-2-n_{ep1}[\zeta(D_1+D_2)+2]+2n_{ep1}}{2n_{ep1}}} \right) \tag{78}$$

When the elliptical asperity is in the completely elastic deformation, the normal contact stiffness $k_{ne}$ can be presented as:

$$k_{ne} = \int_{a_{nec}}^{a_l} k_{ne} n'(a) da = \frac{2f_2(e)}{[f_1(e)]^{1/2} \pi^{1/2}} \frac{\zeta(D_1 + D_2)}{1 - \zeta(D_1 + D_2)} a_l^{\zeta(D_1+D_2)/2} \left( a_l^{\frac{1-\zeta(D_1+D_2)}{2}} - a_{nec}^{\frac{1-\zeta(D_1+D_2)}{2}} \right) \tag{79}$$

## 5. Numerical Analysis and Discussion of Results

### 5.1. Effect of Fractal Parameters and Eccentricity on Contact Area

The above analysis shows that the real contact area is closely related to the fractal dimensions $D$ and the eccentricity $e$ of the elliptical asperities as well as the contact coefficient $\lambda$ for a certain material parameter of the rough surface and normal contact load. If we take the material parameters of the equivalent rough surface as shown in Table 1, the fractal dimension $D_1 = 1.56$, the fractal roughness $G_1 = G_2 = 1.0 \times 10^{-10}$ m, and the value range of fractal dimension $D_2$ is 1.3–1.8. Figure 9 shows the relationship curves between different fractal dimensions $D_2$ and the real contact area of the rough surface.

**Table 1.** Calculated parameters for the mechanical properties of rough surfaces.

| Parameters | Profile 1 | Profile 2 |
|---|---|---|
| Fractal dimension $D$ | 1.3–1.8 | 1.3–1.8 |
| Characteristic scale $G$ (m) | $1.0 \times 10^{-10}$ | $1.0 \times 10^{-10}$ |
| Young's modulus $E$ (Pa) | $2.06 \times 10^{11}$ | |
| Poisson's ratio $\nu$ | 0.26 | |
| Plastic yield stress $\sigma_y$ (Pa) | $235 \times 10^6$ | |

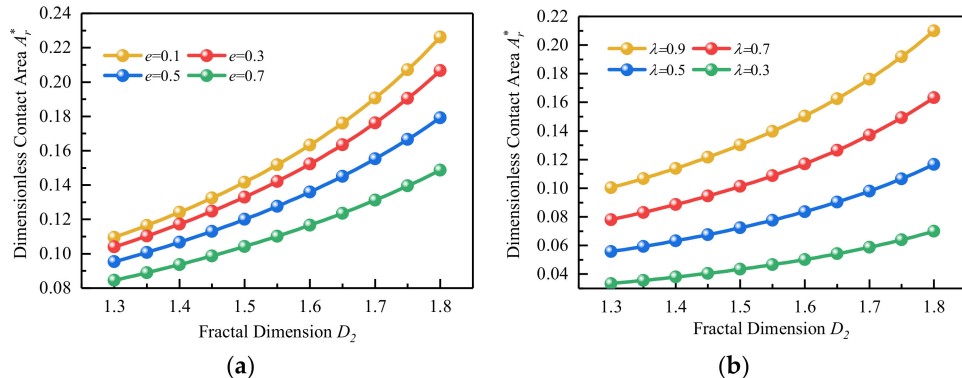

**Figure 9.** Variations of the dimensionless contact area $A_r^*$ with fractal dimension $D$: (**a**) different eccentricity $e$; (**b**) different contact coefficient $\lambda$.

As seen in Figure 9, the real contact area grows in proportion to the fractal dimension. This is because the rough surface becomes flatter as the fractal dimension increases, increasing the real contact area.

As seen in Figure 9a, the real contact area reduces as the eccentricity $e$ increases when the fractal dimensions are determined. This is because the contact area of a single elliptical asperity is less than that of a spherical asperity whenever the eccentricity is considered; so, the real contact area of the rough surface reduces as the eccentricity $e$ increases. As seen in Figure 9b, the real contact area of the rough surface decreases as the contact coefficients grow when the fractal dimensions are given. This is because when the contact coefficients grow, the contact area of the two rough surfaces reduces, resulting in a reduction in the real contact area.

The relationship between the ratio of the elastic contact area to the real contact area and the fractal dimension $D$ at different eccentricities $e$ is shown in Figure 10. As can be demonstrated, although the real contact area decreases as the eccentricity $e$ grows, the ratio of the elastic contact area to the real contact area almost remains constant. The elastic contact area ratio increases as the fractal dimension $D$ increases and stabilizes at 1.55.

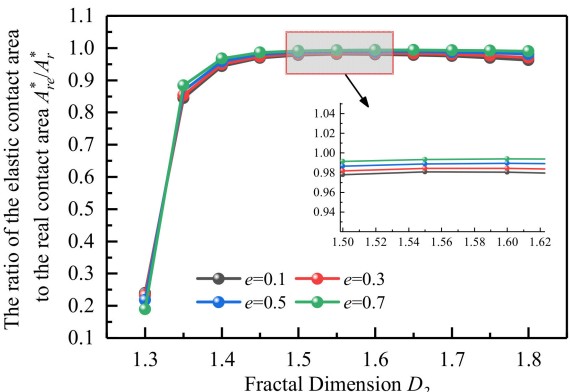

**Figure 10.** Relationship between the ratio of the elastic contact area to the real contact area and fractal dimension $D$ at different eccentricity $e$.

### 5.2. Effect of Fractal Parameters on the Contact Load

The mechanical properties of the model presented in this research are compared to those of the MB contact model in Figure 11. The plastic yield stress of the material $\sigma_y = 235 \times 10^6$ Pa, the elastic modulus $E = 2.06 \times 10^{11}$ Pa, and the Poisson's ratio $\nu = 0.26$. The model in this paper and the revised MB model take the same fractal parameters, namely the fractal dimension $D = D_1 = D_2 = 1.46$, the fractal roughness $G = G_1 = G_2 = 1.0 \times 10^{-10}$ m, and the eccentricity of the model in this paper $e = 0.2$. As can be seen, the mechanical

curves of the model in this paper deviate from those of the MB model, and the real contact load calculated by the model in this paper is less than the real contact load calculated by the MB model at the same real contact area. This is because the eccentricity consequences of the elliptical asperity are accounted for in the calculation model in this paper; hence, the dimensionless contact load of the model in this paper is reduced at the same real contact area.

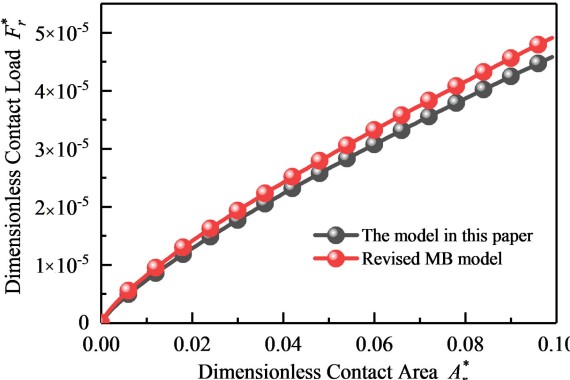

**Figure 11.** Comparison of the mechanical properties between the model in this paper and the MB contact model.

The fractal dimension $D_1 = 1.36$, the fractal roughness $G_1 = G_2 = 1.0 \times 10^{-11}$ m, and we set the fractal dimension $D_2$ as 1.36, 1.46, 1.56, 1.66, and 1.76, respectively. The relationship between the real contact area and the real contact load for different fractal dimensions can be seen in Figure 12a; it can be observed that the real contact load grows as the fractal dimension increases. The reason for this is that the fractal dimension $D$ is positively associated with the rough surface's smoothness, and as the fractal dimension rises, the rough surface's topology performs more finely, decreasing the real contact load on the rough surface.

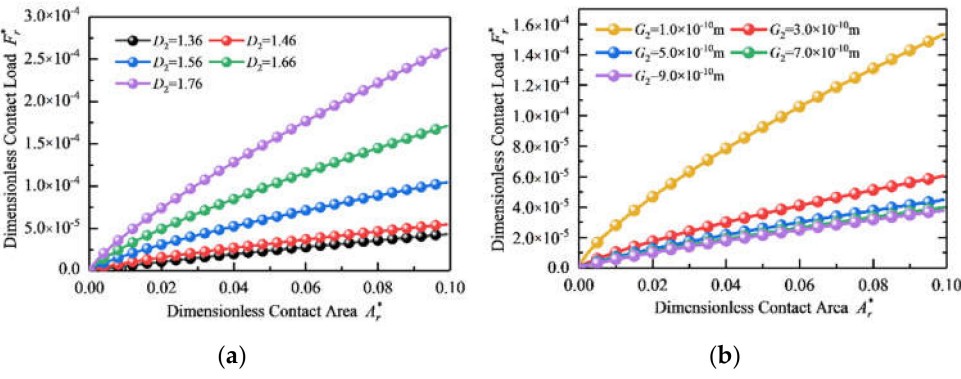

| (a) | (b) |

**Figure 12.** Variations in the dimensionless contact area $A_r^*$ with fractal dimension $D$: (**a**) different fractal dimensions $D$; (**b**) different fractal roughness $G$.

The fractal dimension $D_1 = D_2 = 1.46$, the fractal roughness $G_1 = 1.0 \times 10^{-10}$ m, and we set the fractal roughness $G_2$ as $1.0 \times 10^{-10}$ m, $3.0 \times 10^{-10}$ m, $5.0 \times 10^{-10}$ m, $7.0 \times 10^{-10}$ m, and $9.0 \times 10^{-10}$ m, respectively. Figure 12b illustrates the relationship between the real contact area and the real contact load for various fractal roughness values, demonstrating that the real contact load reduces as the fractal roughness increases. This is because the fractal roughness $G$ is inversely related to the rough surface's smoothness. When the fractal roughness $G$ is raised, the projections and depressions in the rough surface topology increase, resulting in a reduction in the rough surface's real contact load.

The fractal dimension $D_1 = D_2 = 1.46$, the fractal roughness $G_1 = G_2 = 1.0 \times 10^{-11}$ m, and we set the eccentricity $e$ of the elliptical asperity to 0.1, 0.3, 0.5, and 0.7, respectively. The relationship between the real contact area and the real contact load with increasing eccentricity $e$ can be seen in Figure 13a. As seen in the figure, the real contact load reduces as the eccentricity $e$ grows. This is because whenever the eccentricity $e$ grows, the contact area of a single elliptical asperity diminishes, reducing the total contact area of the rough surface and, hence, reducing the rough surface's real contact load.

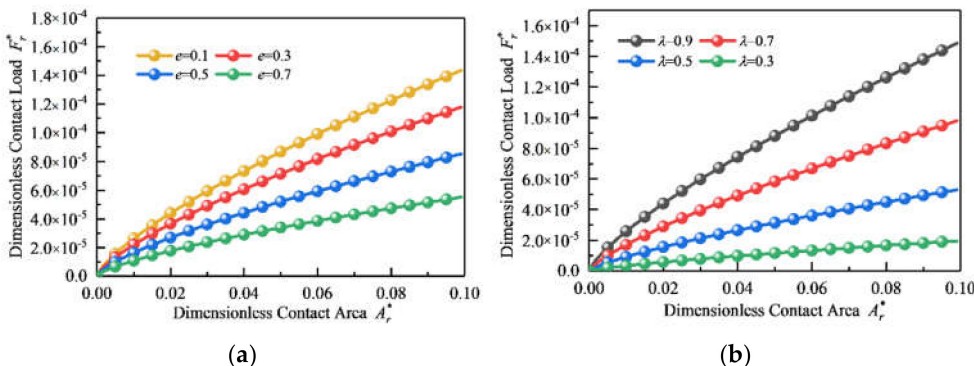

$$(\mathbf{a}) \qquad\qquad\qquad\qquad (\mathbf{b})$$

**Figure 13.** Variations in the dimensionless contact area $A_r^*$ with fractal dimension $D$: (**a**) different eccentricity $e$; (**b**) different contact coefficient $\lambda$.

The fractal dimension $D_1 = D_2 = 1.46$, the fractal roughness $G_1 = G_2 = 1.0 \times 10^{-11}$ m, and we set the contact coefficient to 0.9, 0.7, 0.5, and 0.3, respectively. The relationship between the real contact area and the real contact load for different contact coefficients is shown in Figure 13b. As seen in the figure, the real contact load reduces as the contact coefficients drop. The reason for this is that when the contact coefficients are reduced, the real contact area of the rough surfaces decreases, resulting in a decrease in the real contact load.

### 5.3. Effect of Fractal Parameters on Normal Contact Stiffness

The fractal dimension $D_1 = 1.36$, the fractal roughness $G_1 = G_2 = 1.0 \times 10^{-11}$ m, and we set the fractal dimension $D_2$ as 1.36, 1.46, 1.56, 1.66, and 1.76, respectively. The influence of different fractal dimensions on the relationship between the normal contact stiffness and the real contact area can be seen in Figure 14a. As a consequence, when the real contact area is known, the normal contact stiffness is proportional to the fractal dimension, which suggests that the normal contact stiffness rises as the fractal dimension grows. This is because the fractal dimension does have a physical significance that pertains to the smoothness of the rough surface. On a macroscopic level, the larger the fractal dimension, the higher the roughness value of the rough surface, and the more asperities are present in the contact on the rough surface, enhancing the rough surface's normal contact stiffness.

The fractal dimension $D_1 = D_2 = 1.46$, the fractal roughness $G_1 = 1.0 \times 10^{-10}$ m, and we set the fractal roughness $G_2$ as $1.0 \times 10^{-10}$ m, $3.0 \times 10^{-10}$ m, $5.0 \times 10^{-10}$ m, $7.0 \times 10^{-10}$ m, and $9.0 \times 10^{-10}$ m, respectively. The influence of the specific fractal roughness values on the relationship between the normal contact stiffness and the real contact area is depicted in Figure 14b. As can be observed, the normal contact stiffness is inversely related to the fractal roughness for a given real contact area, namely, the normal contact stiffness decreases as the fractal roughness grows. This is because when the fractal roughness $G$ value improves, the rough surface morphology becomes rougher and the number of asperities in the contact diminishes, decreasing the normal contact stiffness of the rough surface.

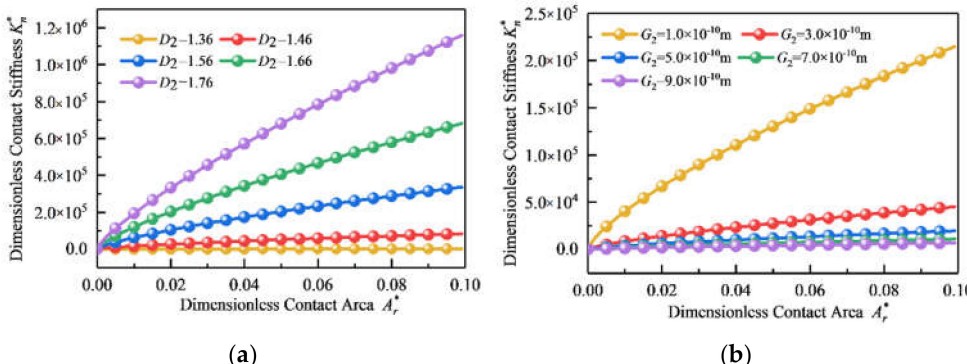

(**a**)　　　　　　　　　　　　　　　　　　　(**b**)

**Figure 14.** Variations in the dimensionless contact stiffness $K_n^*$ with the dimensionless contact area $A_r^*$: (**a**) different fractal dimensions $D$; (**b**) different fractal roughness $G$.

The fractal dimension $D_1 = D_2 = 1.46$, the fractal roughness $G_1 = G_2 = 1.0 \times 10^{-11}$ m, and we set the eccentricity $e$ of the elliptical asperity to 0.1, 0.3, 0.5, and 0.7, respectively. The influence of varying the eccentricity $e$ on the relationship between the normal contact stiffness and the real contact area is illustrated in Figure 15a. As shown in the figure, the normal contact stiffness is inversely proportional to the eccentricity for a given real contact area, which indicates that the normal contact stiffness diminishes as the eccentricity increases. This is because when the eccentricity value grows, the contact area of the single elliptical asperity decreases, resulting in a drop in the rough surface's total actual contact area and consequently a decrease in the rough surface's normal contact stiffness.

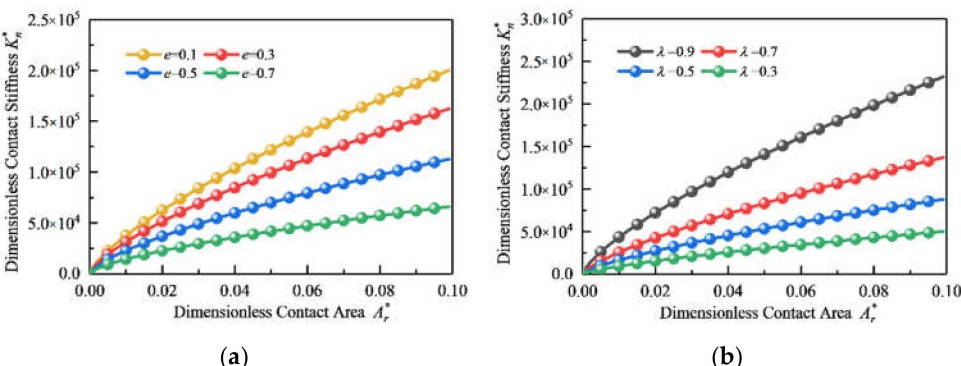

(**a**)　　　　　　　　　　　　　　　　　　　(**b**)

**Figure 15.** Variations in the dimensionless contact stiffness $K_n^*$ with the dimensionless contact area $A_r^*$: (**a**) different eccentricity $e$; (**b**) different contact coefficient $\lambda$.

The fractal dimension $D_1 = D_2 = 1.46$, the fractal roughness $G_1 = G_2 = 1.0 \times 10^{-11}$ m, and we set the contact coefficient to 0.9, 0.7, 0.5, and 0.3, respectively. Figure 15b illustrates the effect of varying the contact coefficients on the relationship between the normal contact stiffness and real contact area. As shown in the graphic, the normal contact stiffness is proportional to the contact coefficients for a given real contact area, which suggests that normal contact stiffness reduces as contact coefficients decrease. This is because the contact coefficients' physical significance is to correct the contact area between two surfaces, and when the integrated curvature of the elastic surface approaches 0, namely, when the two elastic surfaces are in intimate interaction, the contact coefficient value approaches 1 at this time. As the integrated curvature of the elastic surface grows, the contact coefficients and contact area between the elastic surfaces decrease, reducing normal contact stiffness.

## 6. Conclusions

The contact mechanics model for a rough surface with elliptical asperities was established in this paper by comprehensively considering the geometric structure and mechanical properties of elliptical asperities on rough surfaces, which are summarized as follows:

(1) The real contact area of the rough surface is inversely proportional to the fractal dimensions $D_1$ and $D_2$ and the eccentricity $e$ of the elliptical asperities, as well as the contact coefficient. $D$ has a direct influence on the real contact area. The real contact area grows concerning the fractal dimension $D$. Conversely, when the value of fractal dimensions is given, the real contact area declines as the eccentricity $e$ increase, and the real contact area reduces as the contact coefficients decrease;

(2) While the real contact area diminishes as the eccentricity $e$ increases, the ratio of the elastic contact area to the real contact area remains almost constant. The elastic contact area ratio grows as the fractal dimension $D$ increases and stabilizes at 1.55;

(3) The fractal dimension, fractal roughness, eccentricity, and contact coefficient always have an effect on the real contact load and normal contact stiffness of curved surfaces. With the growing fractal dimension and contact coefficient, the real contact load and normal contact stiffness of curved surfaces increase. In comparison, the real contact load and normal contact stiffness diminish as the fractal roughness and eccentricity increase.

**Author Contributions:** Conceptualization, G.Y. and H.M.; methodology, H.M.; software, H.M.; validation, G.Y., H.M. and L.J.; formal analysis, W.L.; investigation, W.L.; resources, L.J.; data curation, H.M.; writing—original draft preparation, H.M.; writing—review and editing, G.Y. and H.M.; visualization, H.M.; supervision, T.V.; project administration, G.Y.; funding acquisition, G.Y. All authors have read and agreed to the published version of the manuscript.

**Funding:** The research was supported by the Major science and technology projects of Heilongjiang Province (Grant No. GA21D004); Heilongjiang Province "hundred million" project science and technology major scientific projects (Grant No. 2019ZX03A03); The Fundamental Research Funds for the Central Universities (Grant No. FRFCU5710052921); Heilongjiang Major Science and Technology Achievement Transformation Project (Grant No. CG21B010); National Key Research and Development Project of China (Grant No. 2019YFE013200).

**Institutional Review Board Statement:** Not applicable.

**Informed Consent Statement:** Not applicable.

**Data Availability Statement:** Not applicable.

**Conflicts of Interest:** The authors declare no conflict of interest.

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
