# Peer review of "Fractal Contact Mechanics Model for the Rough Surface of a Beveloid Gear with Elliptical Asperities"

_applsci, doi:10.3390/app12084071_

Round 1
Reviewer 1 Report
I suggest an improvement of the bibliography, less than 20% are from the last 5 years. Also a re-verification of the formatting (ex line 26 the formula is bold and another dimension, line 481 the title is at the end of the pages)
Author Response
Dear Editors and Reviewers:
Thank you for your letter and for the reviewers’ suggestions concerning our manuscript entitled “Fractal Contact Mechanics Model for Rough Surface of Beveloid Gear with Elliptical Asperities” (ID: applsci-1655256). Those suggestions are all valuable and very helpful for revising and improving our paper, as well as the important guiding significance to our research. We appreciate the Editors/Reviewers’ warm work earnestly and hope that the correction will meet with approval. The main corrections in the paper are as follows:
Point 1: I suggest an improvement of the bibliography, less than 20% are from the last 5 years.
Response 1: Special thanks to you for your detailed review of my manuscript. According to your suggestion, I have improved the bibliography in this paper and added references for the last 5 years related to the research in this paper.
Point 2: Also a re-verification of the formatting.
Response 2: We have recognized our negligence of journal format, and we have corrected all format mistakes in the paper using journal format.

Reviewer 2 Report
This paper presents a fractal contact mechanics model for rough surface of beveloid gear with elliptical asperities. The results obtained are useful for researchers in tribology. Before recommending it for publication, I have some minor comments as follows:
- No comparison with existing results is found in this paper.
- The quality of this paper can be improved if the authors could provide the experimental data for fractal parameters related to the real rough surface of beveloid gear and use them in calculations.
Author Response
Dear Editors and Reviewers:
Thank you for your letter and for the reviewers’ suggestions concerning our manuscript entitled “Fractal Contact Mechanics Model for Rough Surface of Beveloid Gear with Elliptical Asperities” (ID: applsci-1655256). Those suggestions are all valuable and very helpful for revising and improving our paper, as well as the important guiding significance to our research. We appreciate the Editors/Reviewers’ warm work earnestly and hope that the correction will meet with approval. The main corrections in the paper are as follows:
Point 1: No comparison with existing results is found in this paper.
Response 1: Special thanks to you for your detailed review of my manuscript. The comparison with existing results is shown in Figure 11 of this paper, and the comparison results were discussed in detail in this paper.
Point 2: The quality of this paper can be improved if the authors could provide the experimental data for fractal parameters related to the real rough surface of beveloid gear and use them in calculations.
Response 2: Special thanks to you for your good suggestions. We have conducted the roughness measurements of the beveloid gear tooth surface, as seen in Figure 1. And the measured tooth surface topography characteristics were analyzed in this paper.

Reviewer 3 Report
The paper deals with rough contact modelling in beveloid gears. The topic could be surely of interest for scientific and industry community. Nevertheless, in the current state of the article, it is very difficult to detect the novel elements of the proposed model. Moreover, bibliography is very poor compared to the state of the art on this topic. I suggest a strong review of the presentation of model and results, and an extensive editing of English language and style. The paper cannot be accepted in the current state.
Author Response
Dear Editors and Reviewers:
Thank you for your letter and for the reviewers’ suggestions concerning our manuscript entitled “Fractal Contact Mechanics Model for Rough Surface of Beveloid Gear with Elliptical Asperities” (ID: applsci-1655256). Those suggestions are all valuable and very helpful for revising and improving our paper, as well as the important guiding significance to our research. We appreciate the Editors/Reviewers’ warm work earnestly and hope that the correction will meet with approval. The main corrections in the paper are as follows:
Point 1: In the current state of the article, it is very difficult to detect the novel elements of the proposed model.
Response 1: Special thanks to you for your detailed review of my manuscript. The main innovations of this paper are described as follows:
While significant work has been conducted by domestic and foreign scholars to investigate the contact problems between rough surfaces, the current research on the fractal contact model between two rough curved surfaces has generally ignored the influence of surface texture on rough surface contact behavior. However, the topography of rough surfaces varies in the cutting direction and vertical direction, and the contact between two rough surfaces consists of a spectrum of elliptical asperities of varying diameters, with an anisotropic distribution of the elliptical asperities. Therefore, the contact area distribution function of elliptical asperity was proposed for the point contact of curved surfaces by transforming the elastic contact problem between gear meshing surfaces into the contact between elastic curved surfaces with an arbitrary radius of curvature. In addition, a fractal contact mechanics model for the rough surface of a beveloid gear with elliptical asperities was established in this paper, and the influence of tooth surface topography on the contact load and contact stiffness under different fractal parameters was investigated.
Point 2: Moreover, bibliography is very poor compared to the state of the art on this topic. I suggest a strong review of the presentation of model and results.
Response 2: Special thanks to you for your good suggestions. We have re-written the introduction section, and a review of research related to contact analysis of rough surfaces was made in this paper. In the last section of the Introduction, We discuss the problems of the current research on the fractal contact mechanics model for the rough surface, and we emphasized the originality, novelty, and purpose of the research in this paper.
Point 3: An extensive editing of English language and style.
Response 3: This paper has undergone English language editing by MDPI. The text has been checked for correct use of grammar and common technical terms and edited to a level suitable for reporting research in a scholarly journal.

Reviewer 4 Report
In overall, the paper is reasonably good. It provides sufficient details of the proposed model and analysis. It has a proper investigation structure and flow. However, before publishing, the manuscript should be corrected according to the comments and suggestions below:
Section 1. Introduction, Page 1 line 28: An introduction on why contact mechanics models are important to be established. Brief explanation on the usage or application.
Section 1. Introduction, paragraph 1 and 2: Many existing contact models were merely listed and named. However, no proper justification or explanation on why these contact models do not work and the identified gap and motivation to establish the current proposed contact model.
There is a typing error at page 1 line 40, MB instead of M-B.
In-text citation should be before the full stop. E.g., page 1 line 37, …demonstrated [1,2]. Majumdar…
Author Response
Dear Editors and Reviewers:
Thank you for your letter and for the reviewers’ suggestions concerning our manuscript entitled “Fractal Contact Mechanics Model for Rough Surface of Beveloid Gear with Elliptical Asperities” (ID: applsci-1655256). Those suggestions are all valuable and very helpful for revising and improving our paper, as well as the important guiding significance to our research. We appreciate the Editors/Reviewers’ warm work earnestly and hope that the correction will meet with approval. The main corrections in the paper are as follows:
Point 1: Section 1. Introduction, Page 1 line 28: An introduction on why contact mechanics models are important to be established. Brief explanation on the usage or application.
Response 1: Special thanks to you for your detailed review of my manuscript. We have re-written the introduction section according to the Reviewer’s suggestion, and we emphasized the originality, novelty, and purpose of the study in this paper.
Point 2: Section 1. Introduction, paragraph 1 and 2: Many existing contact models were merely listed and named. However, no proper justification or explanation on why these contact models do not work and the identified gap and motivation to establish the current proposed contact model.
Response 2: According to your suggestion, I have improved the Introduction in this paper, and a review of research related to contact analysis of rough surfaces was made in this paper. In the last section of the Introduction, We discuss the problems of the current research on the fractal contact mechanics model for the rough surface, and we emphasized the originality, novelty, and purpose of the study in this paper.
Point 3: There is a typing error at page 1 line 40, MB instead of M-B.
Response 3: We have recognized our incorrect writing, and we have made corrections according to the Reviewer's suggestion.
Point 4: In-text citation should be before the full stop. E.g., page 1 line 37, …demonstrated [1,2]. Majumdar…
Response 4: We have recognized our incorrect writing, and we have made corrections according to the Reviewer's suggestion.

Round 2
Reviewer 2 Report
This paper presents a fractal contact mechanics model for rough surface of beveloid gear with elliptical asperities。 Although the results are limited to the beveloid gear, the method proposed can be extended to other types of machined surfaces. Thus, I recommend it for publication in this journal.
Reviewer 3 Report
Paper has been consistently improved with respect the previous review. Thus, at this state, its publication can be recommended.